# Air Quality and Climate Change, Topic 3 of the Model Inter-Comparison Study for Asia Phase III (MICS-Asia III), Part II: aerosol radiative effects and aerosol feedbacks

Meng Gao[1,2], Zhiwei Han[3,4], Zhining Tao[5,6], Jiawei Li[3,4], Jeong-Eon Kang[7], Kan Huang[8], Xinyi Dong[9], Bingliang Zhuang[10], Shu Li[10], Baozhu Ge[11], Qizhong Wu[12], Hyo-Jung Lee[7], Cheol-Hee Kim[7], Joshua S. Fu[9], Tijian Wang[10], Mian Chin[6], Meng Li[13], Jung-Hun Woo[14], Qiang Zhang[15], Yafang Cheng[13], Zifa Wang[4,11], Gregory R. Carmichael[16]

Department of Geography, Hong Kong Baptist University, Hong Kong SAR, China
State Key Laboratory of Environmental and Biological Analysis, Hong Kong Baptist University, Hong Kong SAR, China
Key Laboratory of Regional Climate-Environment for Temperate East Asia, Institute of Atmospheric Physics, Chinese Academy of Sciences, Beijing, China
University of Chinese Academy of Sciences, Beijing 100049, China
Universities Space Research Association, Columbia, MD, USA
NASA Goddard Space Flight Center, Greenbelt, MD, USA
Department of Atmospheric Sciences, Pusan National University, Busan, South Korea
Department of Environmental Science and Engineering, Fudan University, Shanghai, China
Department of Civil and Environmental Engineering, University of Tennessee, Knoxville, TN, USA
School of Atmospheric Sciences, Nanjing University, Nanjing, China
State Key Laboratory of Atmospheric Boundary Layer Physics and Atmospheric Chemistry, Institute of Atmospheric Physics, Chinese Academy of Sciences, Beijing, China
College of Global Change and Earth System Science, Beijing Normal University, Beijing, China
Multiphase Chemistry Department, Max Planck Institute for Chemistry, Mainz, Germany
Department of Advanced Technology Fusion, Konkuk University, Seoul, South Korea
Ministry of Education Key Laboratory for Earth System Modeling, Center for Earth System Science, Tsinghua 15 University, Beijing, China
Center for Global and Regional Environmental Research, University of Iowa, Iowa City, IA, USA
Correspondence to: M. Gao (mmgao2@hkbu.edu.hk), Z. Han (hzw@mail.iap.ac.cn), and G. R. Carmichael (gcarmich@engineering.uiowa.edu)

## Abstract

Topic 3 of the Model Inter-Comparison Study for Asia (MICS-Asia) Phase III examines how online coupled air quality models perform in simulating wintertime haze events in the North China Plain region and evaluates the importance of aerosol radiative feedbacks. This paper discusses the estimates of aerosol radiative forcing, aerosol feedbacks, and possible causes for the differences among the participating models. Over the Beijing-Tianjin-Hebei (BTH) region, the ensemble mean of estimated aerosol direct radiative forcing (ADRF) at the top of atmosphere, inside the atmosphere and at the surface are -1.1, 7.7 and -8.8 $W/m^2$ during January 2010, respectively. Subdivisions of direct and indirect aerosol radiative forcing confirm the dominant role of direct forcing. During severe haze days (January 17-19, 2010), the averaged reduction in near surface temperature for the BTH region can reach 0.3-1.6 ℃. The responses of wind speeds at 10 m (WS10) inferred from different models show consistent declines in eastern China. For the BTH region, aerosol-radiation feedback induced daytime changes in $PM_{2.5}$ concentrations during severe haze days range from 6.0 to 12.9 $\mu g/m^3$ ($< 6\%$). Sensitivity simulations indicate the important effect of aerosol mixing states on the estimates of ADRF and aerosol feedbacks. Besides, BC exhibits large contribution to atmospheric heating and feedbacks although it accounts for a small share of mass concentration of $PM_{2.5}$.

## 1 Introduction

Aerosols change weather and climate via the following pathways: they absorb and scatter solar and thermal radiation to alter the radiative balance of the earth-atmosphere system (*Gao et al., 2019b; Liu et al., 2011; Jia et al., 2018*), which is referred to as direct effects; and, they serve as cloud condensation nuclei (CCN) and/or ice nuclei (IN) to modify cloud properties, which is referred to as indirect effects (*Haywood and Boucher, 2000*). The suppression of cloud convection induced by direct effects of absorbing aerosols is known as the semi-direct effect (*Huang et al., 2006; Lohmann and Feichter, 2005*). Increases in cloud droplet number can increase cloud albedo for a constant liquid water path (LWP), which is further classified as the first indirect effect or Twomey effect (*Twomey, 1991*). More but smaller cloud droplets reduce precipitation intensity but increase cloud lifetime, which is known as the cloud lifetime or

second indirect aerosol effect (*Albrecht, 1989*). In turn, changes in the radiative balance can alter meteorological variables (e.g. temperature, relative humidity, photolysis rate, etc.) and further the transport, diffusion and chemical conversion of trace gases and aerosols, while changes in clouds can affect in-cloud aqueous-phase chemistry and wet deposition of gases and aerosols.

The impacts of meteorology on chemistry have been explicitly treated in chemical transport models (CTMs). For example, temperature modulates chemical reaction and photolysis rates, affects volatility of chemical species, and biogenic emissions, wind speed and direction determine transport and mixing, and precipitation influences wet deposition (*Baklanov et al., 2014*). However, due to the complexity of these processes and lack of computational resources, the influences of atmospheric compositions on weather and climate have been generally ignored in previous CTMs (*Baklanov et al., 2014*). Studies examining how aerosols interact with weather/climate remain uncertain and limited. Recently, with the rapid development of coupled meteorology and chemistry models, many new studies have been conducted to investigate the aerosol direct and indirect effects and feedbacks (*Baklanov et al., 2017; Forkel et al., 2015; Gao et al., 2016, 2017; Grell et al., 2005; Han et al., 2010; Huang et al., 2016; Jacobson et al., 2007; Saide et al., 2012; Wang et al., 2014; Yang et al., 2011; Zhang et al., 2010*). In highly polluted regions like Asia, aerosol feedbacks can be particularly important (*Gao et al., 2016, 2017*). High concentrations of aerosols would enhance the stability of boundary layer due to reductions in radiation that reach the surface, which in turn can cause further increases in $PM_{2.5}$ concentrations (*Ding et al., 2016; Gao et al., 2016*).

Aerosol feedbacks during haze events in China have been explored using multiple online coupled meteorology-chemistry models, including WRF-Chem (the Weather Research Forecasting model coupled with Chemistry, *Chen et al., 2013, 2018; Gao et al., 2016, 2017, 2019a; Liu et al., 2015*), WRF-CMAQ (Community Multiscale Air Quality, *Wang et al., 2014*). Nevertheless, large uncertainties remain in the modelling of these processes, due to the lack of direct observational constraints and challenges in predicting properties of aerosols. Thus, the inter-comparison of coupled meteorology-chemistry models is of great significance to better understand the differences, causes, and uncertainties within these processes.

Topic 3: air quality and climate change within the Model Inter-Comparison Study for Asia

Phase III (MICS-Asia phase III) was initialized to address these issues (*Gao et al., 2018a*). Results from seven applications of fully online coupled meteorology-chemistry models using harmonized emission and chemical boundary conditions were submitted to this topic (*Gao et al., 2018a*). These model applications include two applications of WRF-Chem by different institutions, two applications of the National Aeronautics and Space Administration (NASA) Unified WRF (NU-WRF) model with different model resolutions, one application of the Regional Integrated Environment Modeling System with Chemistry (RIEMS-Chem, *Han et al., 2010*), one application of the coupled Regional Climate Chemistry Modeling System (RegCCMS), and one application of the coupled WRF-CMAQ model (*Gao et al., 2018a*). More detailed information of the participating models, and information about how the experiments were designed and how models perform have been archived in *Gao et al. (2018a)*.

In this paper, we analyze the results from the participating models to address the following questions: (1) how large is the aerosol radiative forcing during winter haze episodes in China and how differently are models estimating it? (2) how do aerosol feedbacks change meteorological variables? and how do current models differ in estimating these changes? (3) how do aerosol feedbacks contribute to the evolution of high aerosol concentrations during winter haze episodes? and what are the best estimates from different models? And (4) what are the major causes of the differences among the models? Sect. 2 describes briefly how the experiments were designed and how models perform. Sect. 3 presents the estimates of aerosol direct radiative forcing inferred from multiple models, including the separation of direct and indirect effects. In Sect. 4, we discuss the impacts of aerosol-radiation feedbacks on meteorological variables and $PM_{2.5}$ concentrations. Sect. 5 illustrates the sensitivity of aerosol forcing and feedbacks to different processes in the model, and the summary is presented in Sect. 6.

## 2 Overview of MICS-Asia III Topic 3

The participants were requested to use common emissions to simulate air quality during January 2010 and submit requested model variables. The participating models include one application of the Weather Research Forecasting model coupled with Chemistry (WRF-Chem;

Fast et al., 2006; *Grell et al., 2005*) by Pusan National University (PNU) (M1); one application of the WRF-Chem model by the University of Iowa (UIOWA) (M2); two applications (two domains: 45 and 15 km horizontal resolutions) of the National Aeronautics and Space Administration (NASA) Unified WRF (NU-WRF; *Peters-Lidard et al., 2015*) model by the Universities Space Research Association (USRA) and NASA's Goddard Space Flight Center (M3 and M4); one application of the Regional Integrated Environment Modeling System with Chemistry (RIEMS-Chem; *Han et al., 2010*) by the Institute of Atmospheric Physics (IAP), Chinese Academy of Sciences (M5); one application of the coupled Regional Climate Chemistry Modeling System (RegCCMS; *Wang et al., 2010*) from Nanjing University (M6); and one application of the coupled WRF-CMAQ (Community Multiscale Air Quality) model by the University of Tennessee at Knoxville (UTK) (M7) (Table 1). A new Asian emission inventory was developed for MICS-Asia III by integrating state-of-the-art national or regional inventories (*Li et al., 2017*), which was provided to all modeling groups, along with biogenic emissions, biomass burning emissions, etc. Simulations from two global chemical transport models (e.g., GEOS-Chem (The Goddard Earth Observing System Model-Chemistry) and MOZART (Model for OZone And Related chemical Tracers)) were provided as boundary conditions for MICS-Asia III. The entire month of January 2010 was simulated and covered by one single simulation for each participating model. Comprehensive model evaluations indicate that all models could capture the observed near-surface temperature and water vapor mixing ratio, but overestimated near-surface wind speeds. These models were able to represent the observed daily maximum downward shortwave radiation, particularly low values during haze days. The observed variations of air pollutants, including $SO_2$, $NO_x$, CO, $O_3$, $PM_{2.5}$, and $PM_{10}$, were reproduced by these models. However, large differences in the models were found in the predicted $PM_{2.5}$ chemical compositions.

## 3 Aerosol Direct and Indirect Forcing

**Fig. 1** shows the monthly mean all-sky aerosol direct radiative forcing (ADRF) over China. The spatial distributions of ADRF at the surface and inside the atmosphere inferred from multiple models are generally consistent, with the largest values in eastern and southwestern

China. Over the Beijing-Tianjin-Hebei (BTH) region (areas marked in Figure S1), M7 reports
the highest ADRF at the surface (-17.0 W/m$^2$), and the largest ADRF inside the atmosphere
(14.6 W/m$^2$) (**Table 2**). M6 shows the lowest ADRF both at the surface and inside the
atmosphere (-3.6 and 3.6 W/m$^2$) (**Table 2**). It is noticed that M6 predicts lower aerosol optical
depth (AOD) than M7 (*Gao et al., 2018a*), which could partly explain the weaker ADRF
estimated by M6. M6 uses an external assumption of aerosol mixing states, which is likely to
cause weaker absorption and ADRF in the atmosphere (*Conant et al., 2003*). However, the
reported ADRF at the top of the atmosphere (TOA) vary widely, and no consensus is reached
on whether the forcing is positive or negative. The spatial pattern of ADRF at the TOA inferred
from M5 are consistently negative across the modeling domain, while the results inferred from
other models are patchy with positive values to the north or to the southwest (**Fig. 1**). Consistent
negative ADRF at the TOA estimated by M5 is related to the strong negative forcing at the
surface and the predicted high concentrations of sulfate by M5 (*Gao et al., 2018a*). Over the
BTH region, simulated ADRF at the TOA range from -2.6 to 0.2 W/m$^2$ (**Table 2**). *Li et al.*
*(2010)* reported observation-based estimates of aerosol radiative forcing across China to be
0.3±1.6 at the TOA. *Chung et al. (2005)* and *Chung et al. (2010)* estimated the forcing over
south Asia to be -2.9 W/m$^2$ and -3.6 W/m$^2$ at the TOA, respectively. The magnitudes of the
model estimated aerosol radiative forcing values are generally in line with these estimates
inferred from observations, while discrepancies among models could be due to assumptions of
aerosol mixing states and other model treatments (parameterization of hygroscopicity, soil dust,
etc.). The discussions on how different model treatments affect the results of ADRF is provided
in Sect. 5.
**Fig. 2** exhibits the ensemble mean of monthly averaged ADRF at the TOA, inside the
atmosphere and at the surface. Elevated forcing inside the atmosphere and at the surface are
mainly located in east China. However, the ensemble mean of forcing at the TOA over the
ocean is slightly higher than that over the land. Over the BTH region, the ensemble mean of
ADRF at the TOA, inside the atmosphere and at the surface are -1.1, 7.7 and -8.8 W/m$^2$,
respectively. In winter, the aerosol radiative forcing in China is largely contributed by the
power sector and residential sector, but with different signs of the contribution (*Gao et al.,*
*2018b*).
M4 and M5 further provide subdivision of direct and indirect aerosol radiative forcing. As
listed in **Table 3**, although the magnitudes of forcing estimated by M4 and M5 differ from each
other, the dominant roles of direct forcing are consistent. Over North China and during
wintertime, aerosol indirect forcing is negligible due to the lack of water vapor and the stable
weather conditions.

**4 Impact of aerosol feedbacks on meteorological variables and PM$_{2.5}$**
**concentrations**
When extreme haze events happen, high aerosol loadings can reduce significantly the
shortwave radiation reaching the surface, modifying near-surface temperature (*Gao et al.,*
*2017*). **Fig. 3** displays the aerosol-radiation feedback induced changes in temperature at 2 m
(T2) from M1 (a), M2 (b), M4 (c), M5 (d), M6 (e), M7 (f) (Table 1: M1: WRF-Chem, Pusan
National University; M2: WRF-Chem, University of Iowa; M4: NU-WRF, NASA; M5:
RIEMS-Chem, Institute of Atmospheric Physics; M6: RegCCMS, Nanjing University; M7:
WRF-CMAQ, University of Tennessee; *Gao et al., 2018a*). The participating models show
different degrees of reductions in T2. M5 exhibits the most widespread areas with reductions,
which include Northeastern China. However, significant reductions in T2 inferred from other
models are mainly concentrated in southern and eastern China (**Fig. 3**). In Beijing (areas
marked in Figure S1), the monthly averaged reductions in T2 from multiple models range from
0 to 0.7 ℃, with the greatest changes calculated from M4 (**Table 2**). In the Beijing-Tianjin-
Hebei (BTH) region, similar magnitudes (0-0.8 ℃) are found. When only severe haze days
(January 17-19) are considered, the averaged reductions in T2 for Beijing (0.1-1.7 ℃) and the
BTH region (0.3-1.6 ℃) are further enhanced (**Table 4**). In terms of aerosol-radiation feedback
induced temperature reduction, M1 and M2 generally report similar magnitudes, which are
lower than M4, M5 and M7. Model evaluations of PM$_{2.5}$ composition in *Gao et al. (2018a)*
reveal that M4 overpredicts the concentrations of organic carbon, which could be one of the
reasons for the higher estimated reductions in T2 due to aerosols.
Pronounced decreases in water vapor at 2 m (Q2) are mostly located in southern China (**Fig.**
**4**), where water vapor is more abundant due to the proximity to the sea. During extreme haze

days, the aerosol-radiation feedback induced decreases in Q2 in the BTH region from multiple models range from 0.07 to 0.29 g/kg, with the lowest estimate from M1 and the highest from M4 (**Table 4**).

The responses of wind speeds at 10 m (WS10) inferred from different models are generally consistent, displaying decreases in eastern China except M6 (**Fig. 5**). In the BTH region, the monthly mean aerosol-radiation feedback induced decreases in WS10 range from 0.02 to 0.09 m/s (**Table 2**), and more pronounced reductions are suggested by M4 and M7 (**Fig. 5**).

Because of aerosol-radiation feedback, most models report that surface $PM_{2.5}$ concentrations are enhanced in China, with the exception of M6 (**Fig. 6**). It is also noteworthy that $PM_{2.5}$ concentrations decrease in the Gobi desert and Taklimakan desert of western China in M5 and M2, which is caused by the decreased wind speed near the surface due to the weakened downward transport of momentum from upper layer above boundary layer to the surface (*Han et al., 2013*). The changes of $PM_{2.5}$ concentrations suggested by M6 are patchy over east China, with decreases to the north and to the southwest. The monthly mean $PM_{2.5}$ are enhanced by 0.1-1.4 µg/m³ for Beijing, and by 0.8-2.2 µg/m³ for the BTH region. The enhancement fractions are generally below 2% for Beijing, and below 4% for the BTH region (**Table 2**).

To further understand how aerosol-radiation feedback contributes to the formation of haze event, we calculate the mean increase during extreme haze days (January 17-19). For the BTH region, the contribution of aerosol-radiation feedback to $PM_{2.5}$ concentrations are lower than 4%, and the enhancement are below 8.5 µg/m³. *Gao et al. (2017)* demonstrates that the aerosol-radiation feedback induced changes in $PM_{2.5}$ are negligible during nighttime, so we further calculate daytime mean changes, as listed in **Table 4**. For the BTH region, M2 reports the largest enhancement (12.9 µg/m³) of $PM_{2.5}$ concentrations during daytime. Other models, except M6, report similar magnitudes of enhancement, ranging from 5.3 to 6.6 µg/m³. The enhancement fraction remains less than 6% for the BTH region, and below 8.3% for Beijing. **Table 4** also displays the maximum enhancement of $PM_{2.5}$ during haze days overthe BTH region. M7 suggests the largest $PM_{2.5}$ enhancement (up to 60.9 µg/m³), followed by M2 (up to 55.4 µg/m³). Other three models, M1, M4, M5, and M6 indicate the aerosol-radiation induced increase in $PM_{2.5}$ can reach up to more than 20 µg/m³ in the BTH region (**Table 4**).

The contributions of aerosol-radiation feedback to haze formation in China have been

investigated in many previous studies (*Ding et al., 2016; Gao et al., 2015; Gao et al., 2016; Liu et al., 2018; Wang et al., 2014a; Wang et al., 2014b; Wang et al., 2015; Wu et al., 2019; Zhang et al., 2015; Zhang et al., 2018; Zhong et al., 2018*), but the reported values diverge. *Ding et al. (2016)*, *Wang et al. (2014a)* and *Zhong et al. (2018)* indicate that the aerosol radiative effects can increase $PM_{2.5}$ by more than 100 µg/m$^3$ (maximum hourly changes) or +70%. *Gao et al. (2015), Wang et al. (2014b), Wang et al. (2015),* and *Zhang et al. (2018)* suggest that the contributions are generally within the range of 10-30%. These reports are different from this study in terms of study periods, region, and pollution levels. Most of previous reports focused on the January 2013 haze episodes (*Wang et al., 2014a*), while the monthly mean concentrations of $PM_{2.5}$ in January 2010 are nearly 50% lower than that of January 2013. According to the findings in this study, the contribution of aerosol-radiation feedback to haze formation during January 2010 are generally below 10%. Uncertainties still remain as suggested by the errors in the simulated chemical compositions (*Gao et al., 2018a*). Concentrations of sulfate and organic aerosol are generally underestimated by most of the participating models, and M4 overestimates the concentrations of organic aerosols (*Gao et al., 2018a*). These model errors were attributed to the incomplete multiphase oxidation mechanisms of sulfate, and different treatments of secondary organic aerosol (SOA) formation in these models (*Gao et al., 2018a*).

## 5 Sensitivity to Different Processes

To explore the potential causes for the differences among models, and the major factors that influence aerosol-radiation feedback, several sensitivity simulations were conducted with the RIEMS-Chem model (M5) (*Han et al., 2010*). These simulations aim to examine the effects of mixing states of aerosols, hygroscopic growth, black carbon and soil dust.

5.1 Aerosol mixing states

In the control simulation, inorganic aerosols and BC are assumed to be internally mixed to form a homogeneous mixture. The refractive index of this mixture is estimated using the volume-weighted average of the refractive index of individual component. The size of the mixture is prescribed to be the maximum size of the mixed aerosol components. For example, the size of

the mixture of sulfate and BC is set to be equal to the size of sulfate, assuming a small BC
particle sticking to a larger sulfate particle.
An additional simulation was conducted with the aerosols were treated as externally mixed,
and the corresponding results are displayed in **Fig. 7-9**. For external mixing assumption, each
aerosol component is considered individually, and the total AOD is calculated as the sum of
extinction by each aerosol component. Compared with the results with internal mixing
assumption, results with external mixing assumption generally exhibit a weaker (negative)
ADRF at the surface (~15%), a stronger (negative) ADRF at TOA (~50%) and a decreased
(positive) ADRF in the atmosphere (~30%) (**Fig. 9a, 9f, 9k**). These responses of ADRF to the
assumption of aerosol mixing states are consistent with *Conant et al. (2003)*. However, *Curci*
*et al. (2015)* reported lower AOD with internal mixing assumption than with external mixing
assumption. In *Curci et al. (2015),* aerosol mass was distributed in less numerous particles with
larger sizes. As a result, fewer scattering agents and lower AOD were estimated.
Aerosol feedbacks estimated by M5 also tend to be weaker with external mixing assumption
than with internal mixing assumption (changes in surface meteorological variables and $PM_{2.5}$
concentrations, **Fig. 8a, 8d, 8g, and 8j**). The monthly averaged changes in T2, WS10 and $PM_{2.5}$
are -0.6 °C, -0.04 m/s and 2.2 μg/m$^3$ for the BTH region with internal mixing assumption, while
the corresponding values change to -0.6 °C, -0.03 m/s and 1.8 μg/m$^3$ with external mixing
assumption. These differences emphasize the important influences of aerosol mixing states on
the estimates of ADRF and aerosol feedbacks. However, aerosol mixing states are also varying
with time and location. Measurements in North China suggest that aerosols are partially
internally mixed, and the fraction of internal mixing increased from clean to haze periods (*Li*
*et al., 2014*).

5.2 Hygroscopic growth
Given the appreciable effect of aerosol hygroscopic growth on ADRF (*Li et al., 2014*), another
simulation was conducted with decreased relative humidity (RH). In this simulation, FNL
nudging was applied above boundary layer to reduce RH This perturbation of RH was based
on the fact that M5 overestimates relative humidity (water vapor mixing ratio) (*Gao et al.,*
*2017*). With FNL nudging, RH was reduced by 5-10% in the BTH and by ~25% in the middle
and lower reaches of the Yangtze River, leading to lower values of AOD (**Fig. 7f**) and weaker
ADRF at the surface and TOA (**Fig. 9e, 9j, and 9o**, about 10% lower in the BTH region).

5.3 Soil dust and sea salt
M5 (RIEMS-Chem) includes naturally emitted soil dust and sea salt, while the other models
except M2 (WRF-Chem, University of Iowa) do not consider soil dust in their model settings.
In an additional sensitivity simulation, soil dust and sea salt emissions were turned off in M5
to examine the influence on ADRF and aerosol feedbacks (**Fig. 9d, 9l and 9n**). In January 2010,
significant amounts of soil dust were emitted from the Taklimakan desert, influencing wide
areas of China. M5 estimates that the monthly mean ADRF at the surface due to dust and sea
salt is about -12 $W/m^2$ over the Taklimakan desert, -4~-6 $W/m^2$ in the middle reaches of the
Yellow River and the Yangtze River Delta, and about -2~-4 $W/m^2$ over the BTH region. Over
the BTH region, the contribution of dust and sea salt aerosols to total ADRF can reach 5~10%.
Table 2 illustrates that M5 predicts the largest (negative) radiative forcing at TOA over the BTH
region. The above analyses with reduced relative humidity and perturbations in dust and sea
salt suggest that the inclusion of dust and overprediction of relative humidity by M5 are
important reasons.

5.4 The effect of BC
Two sets of simulations, namely without BC and with doubled BC concentrations, were
conducted to examine the influences of BC on aerosol radiative forcing and feedbacks. In the
control simulation, the aerosol induced changes in monthly T2, WS10 and $PM_{2.5}$ are -0.6 °C, -
0.04 m/s and 2.2 μg/$m^3$ for the BTH region, respectively. When BC is not included (only
scattering aerosols and dust), the corresponding aerosol induced changes are -0.5 °C, -0.02 m/s
and 1.0 μg/$m^3$, respectively. When BC concentrations are doubled, these values change to -0.7
°C, -0.05 m/s and 3.2 μg/$m^3$, respectively. The comparison between the control case and two
additional sensitivity cases indicates that the changes caused by BC are comparable to those by
scattering aerosols. The contribution of BC to aerosol feedbacks can reach up to 40~50%. It is
also found that the influence of BC on aerosol feedbacks with internal mixing assumption is
larger than that with external mixing assumption (Figure not shown).

Large uncertainties still remain in the estimates of the role of BC in aerosol feedbacks relative to scattering aerosols. *Gao et al. (2016)* suggested that the impacts of BC on boundary layer height and $PM_{2.5}$ concentrations can account for as high as 60% of the total aerosol feedbacks in the North China Plain at 2 p.m., although it only accounts for a small share of PM in terms of mass concentration. *Qiu et al. (2017)* indicated that $PM_{2.5}$ concentrations averaged over the North China Plain increased by 16.8% and 1.0% due to scattering aerosols and BC, respectively. It should be noted that most participating models, including RIEMS-Chem, tend to underpredict the total mass concentrations of scattering aerosols (inorganic and organic aerosols) by up to a factor of two over the study period, leading to overestimation of the contribution of BC.

## 6 Summary

Topic 3 of MICS-Asia III (*Gao et al., 2018a*) focuses on understanding how current online coupled air quality models perform in capturing extreme aerosol pollution event in North China and how aerosols interact with radiation and weather. Seven applications of different online coupled meteorology-chemistry models were involved in this activity. *Gao et al. (2018a)* has demonstrated that main features of the accumulation of air pollutants are generally well represented, while large differences in the models were found in the predicted $PM_{2.5}$ chemical compositions. These inconsistencies would lead to differences in estimated ADRF and aerosol feedbacks.

The spatial distributions of ADRF at the surface and inside the atmosphere inferred from multiple models are generally consistent, but the spatial distributions of ADRF at the TOA estimated by these models greatly differ. Over the BTH region, the ensemble mean of ADRF at the TOA, inside the atmosphere and at the surface are -1.1, 7.7 and -8.8 $W/m^2$, respectively. Subdivisions of direct and indirect aerosol radiative forcing confirm the dominant roles of direct forcing.

During severe haze days (January 17-19), the averaged reduction in T2 for the BTH region can reach 0.3-1.6 ℃. The responses of wind speeds at 10 m (WS10) inferred from different models show consistent declines in eastern China. For the BTH region, aerosol-radiation feedback

induced changes in daytime PM$_{2.5}$ range from 5.3 to 12.9 µg/m$^3$ (< 6%). Our findings differ
from previous studies (*Ding et al., 2016; Gao et al., 2015; Gao et al., 2016; Liu et al., 2018;*
*Wang et al., 2014a; Wang et al., 2014b; Wang et al., 2015; Wu et al., 2019; Zhang et al., 2015;*
*Zhang et al., 2018; Zhong et al., 2018*) in terms of study period, region and pollution levels.
The monthly mean concentrations of PM$_{2.5}$ in January 2010 (current study period) are about
50% lower than those in January 2013.
Sensitivity simulations were conducted with the RIEMS-Chem model (M5) to understand the
influences of aerosols mixing states, hygroscopic growth, black carbon and soil dust. The
results indicate the important effect of aerosol mixing states on the estimates of ADRF and
aerosol feedbacks. It was also found that BC exhibits large contribution to atmospheric heating
and feedbacks, but uncertainties remain in estimating its contribution given the fact that the
observed aerosol chemical components were not perfectly simulated. *Huang et al. (2015)*
separated the contributions of different aerosol components to aerosol direct radiative forcing,
highlighting the roles of BC and sulfate. Future studies are also needed to improve predicitons
of aerosol chemical components and to separate the effects of individual aerosol component on
aerosol feedbacks.

**Author Contributions**

M.G., Z.H., and G.R.C. designed the study, and M.G. processed and analyzed the data. M.G.,
Z.H., and G.R.C. wrote the paper with inputs from all other authors.

**Data availability**

The measurements and model simulations data can be accessed through contacting the
corresponding authors.

**Competing interests**

The authors declare that they have no conflict of interests.

**Acknowledgement**

The authors would like to acknowledge support for this project from the National Natural

Science Foundation of China (91644217 and 41620104008). This work was supported also by the special fund of State Key Joint Laboratory of Environment Simulation and Pollution Control (19K03ESPCT), Natural Science Foundation of Guangdong Province (2019A1515011633), and National Natural Science Foundation of China (NSFC91543202).

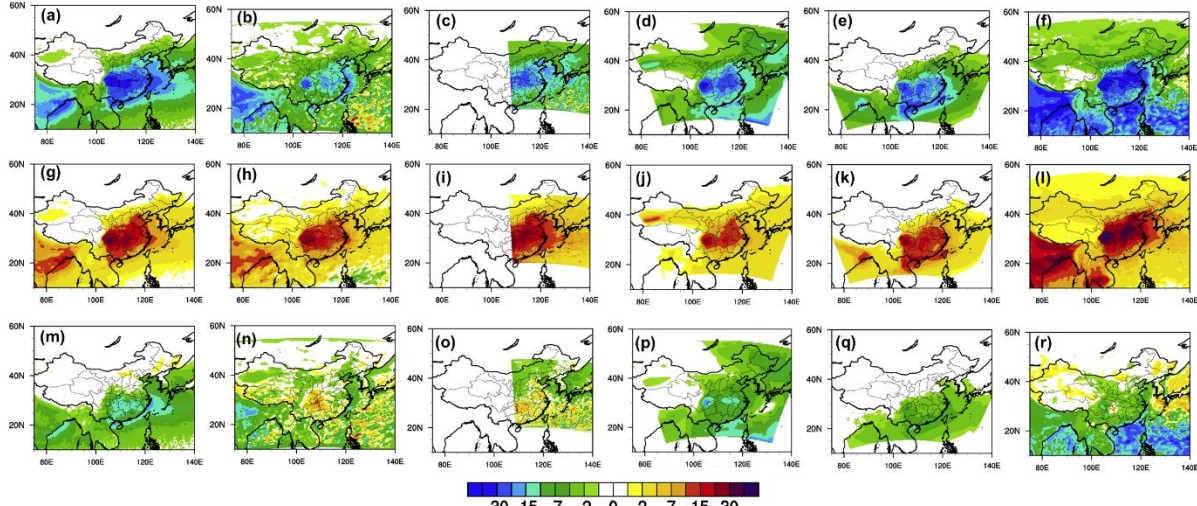

Figure 1. Monthly (January 2010) mean aerosol direct radiative forcing at the surface, inside the atmosphere and at the top of the atmosphere inferred from M1 (a, g, m), M2 (b, h, n), M4 (c, i, o), M5 (d, j, p), M6 (e, k, q), M7 (f, l, r) (M1: WRF-Chem, Pusan National University; M2: WRF-Chem, University of Iowa; M4: NU-WRF, NASA; M5: RIEMS-Chem, Institute of Atmospheric Physics; M6: RegCCMS, Nanjing University; M7: WRF-CMAQ, University of Tennessee; *Gao et al., 2018a*)

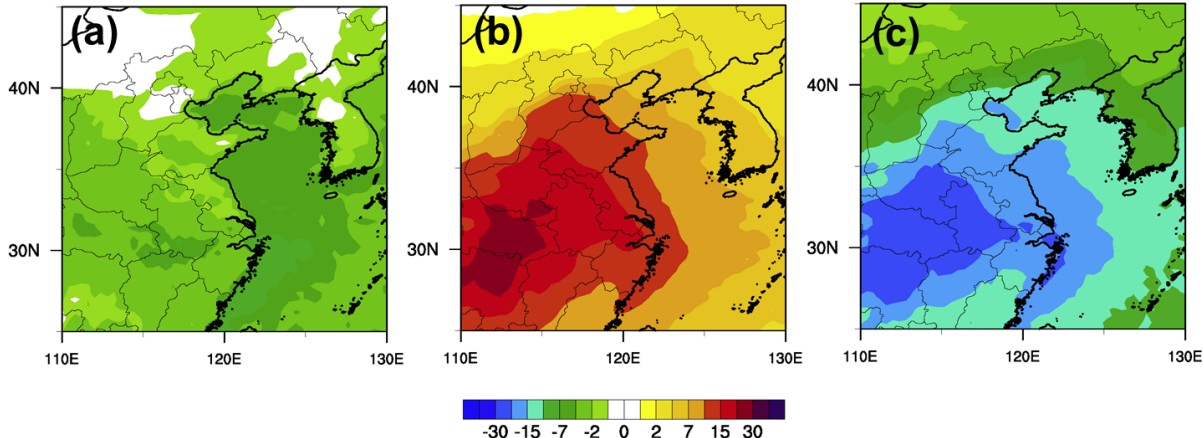

Figure 2. Ensemble mean of monthly (January 2010) mean aerosol direct radiative forcing at the top of the
412                           atmosphere (a), inside the atmosphere (b) and at the surface (c)




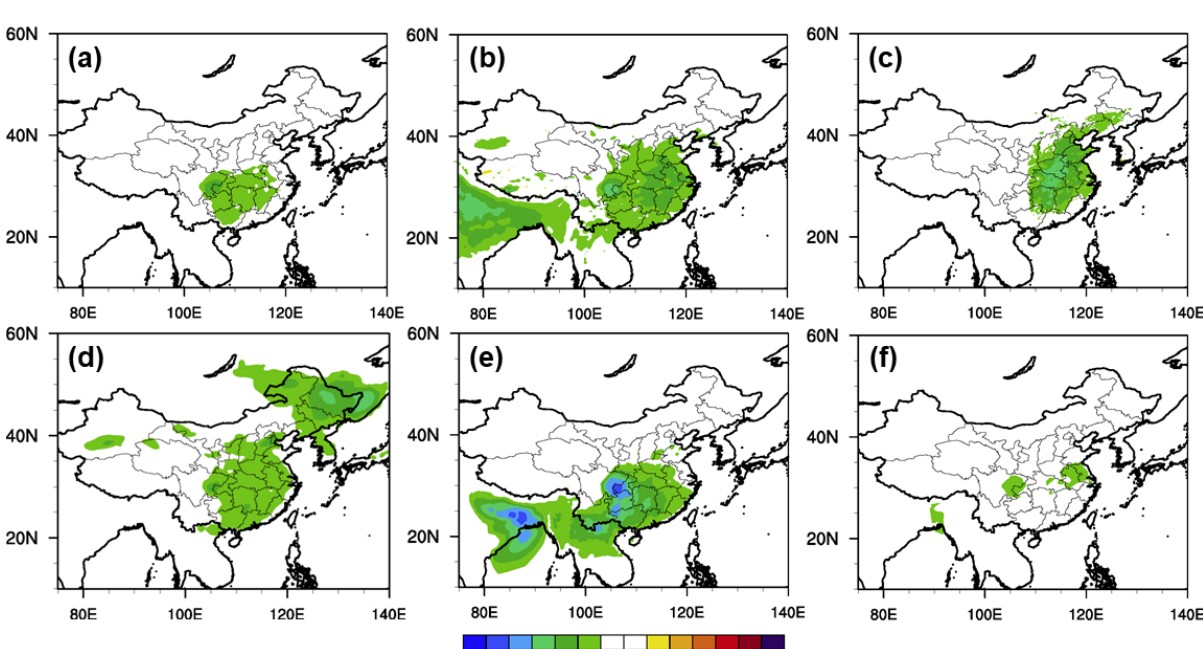

Figure 3. Monthly (January 2010) mean changes in temperature at 2 m (T2, ºC) due to aerosol radiative effects
from M1 (a), M2 (b), M4 (c), M5 (d), M6 (e), M7 (f) (M1: Pusan National University; M2: University of Iowa;
M4: NASA; M5: Institute of Atmospheric Physics; M6: Nanjing University; M7: University of Tennessee; *Gao*
420                                          *et al., 2018a*)


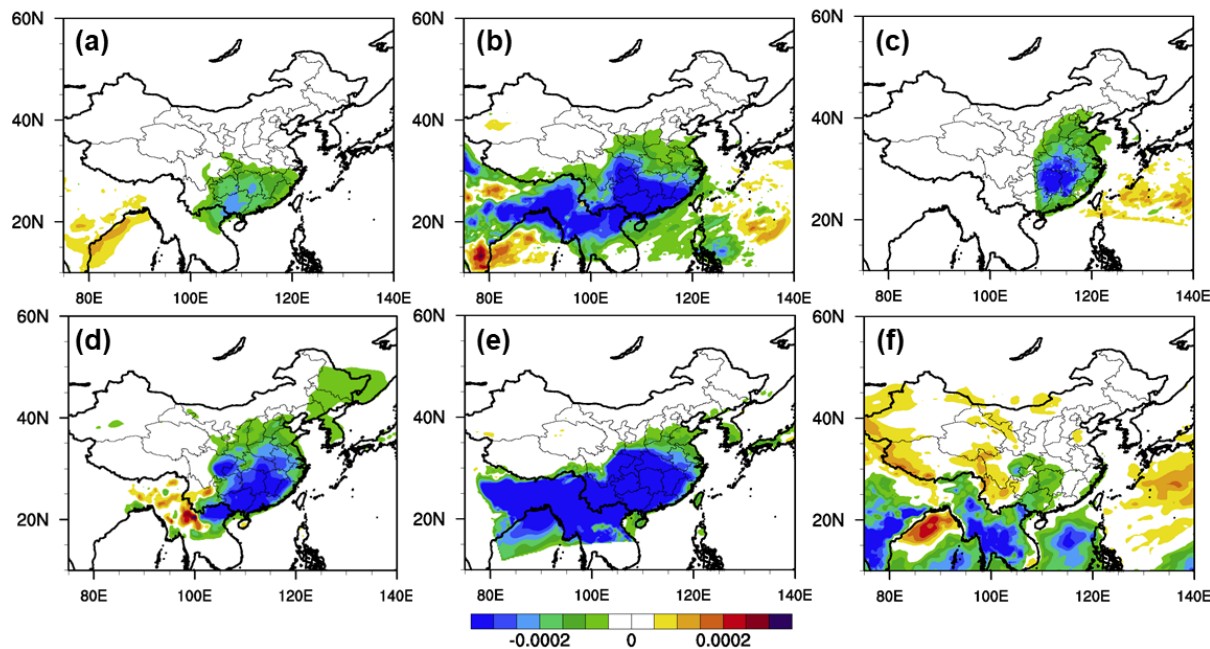


Figure 4. Monthly (January 2010) mean changes in water vapor at 2 m (Q2, kg/kg) due to aerosol radiative
effects from M1 (a), M2 (b), M4 (c), M5 (d), M6 (e), M7 (f) (M1: Pusan National University; M2: University of
Iowa; M4: NASA; M5: Institute of Atmospheric Physics; M6: Nanjing University; M7: University of
Tennessee; *Gao et al., 2018a*)

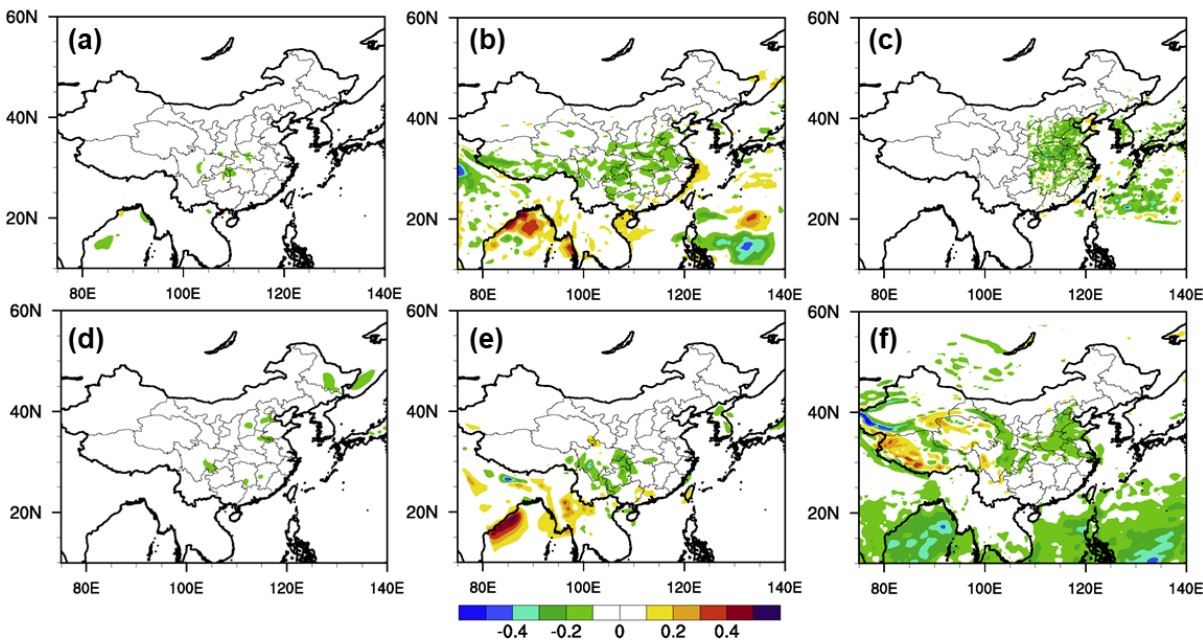


Figure 5. Monthly (January 2010) mean changes in wind speeds at 10 m (WS10, m/s) due to aerosol radiative
effects from M1 (a), M2 (b), M4 (c), M5 (d), M6 (e), M7 (f) (M1: Pusan National University; M2: University of
Iowa; M4: NASA; M5: Institute of Atmospheric Physics; M6: Nanjing University; M7: University of
Tennessee; *Gao et al., 2018a*)

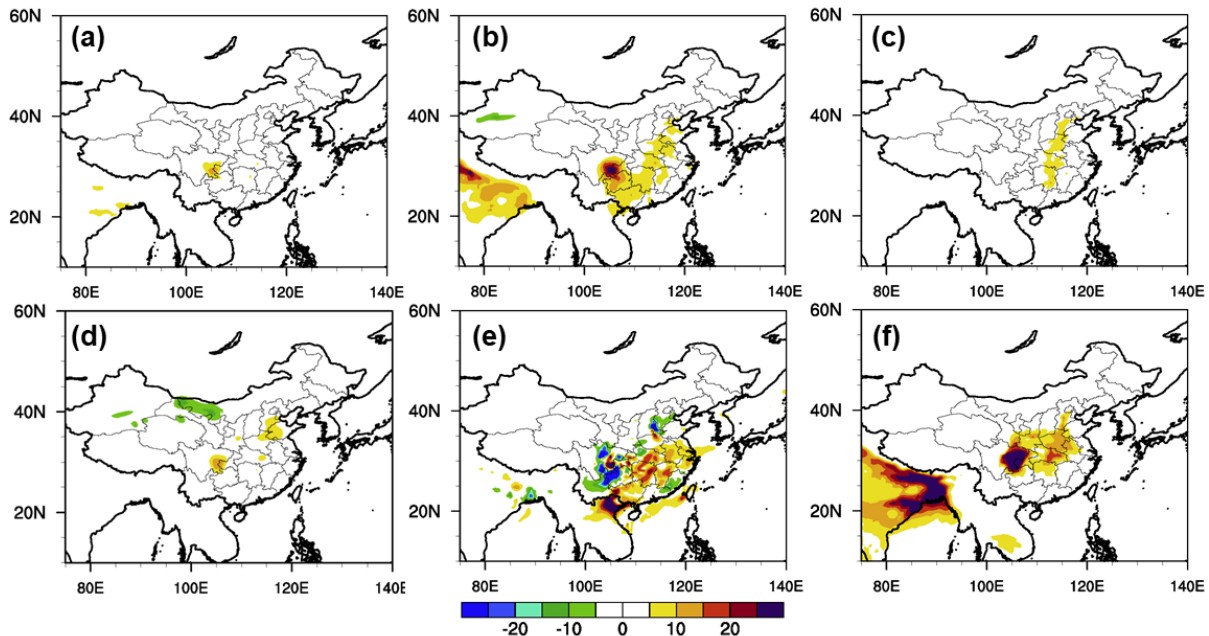

Figure 6. Monthly (January 2010) mean changes in surface PM$_{2.5}$ (µg/m$^3$) due to aerosol radiative effects from
M1 (a), M2 (b), M4 (c), M5 (d), M6 (e), M7 (f) (M1: Pusan National University; M2: University of Iowa; M4:
NASA; M5: Institute of Atmospheric Physics; M6: Nanjing University; M7: University of Tennessee; *Gao et*
*al., 2018a*)

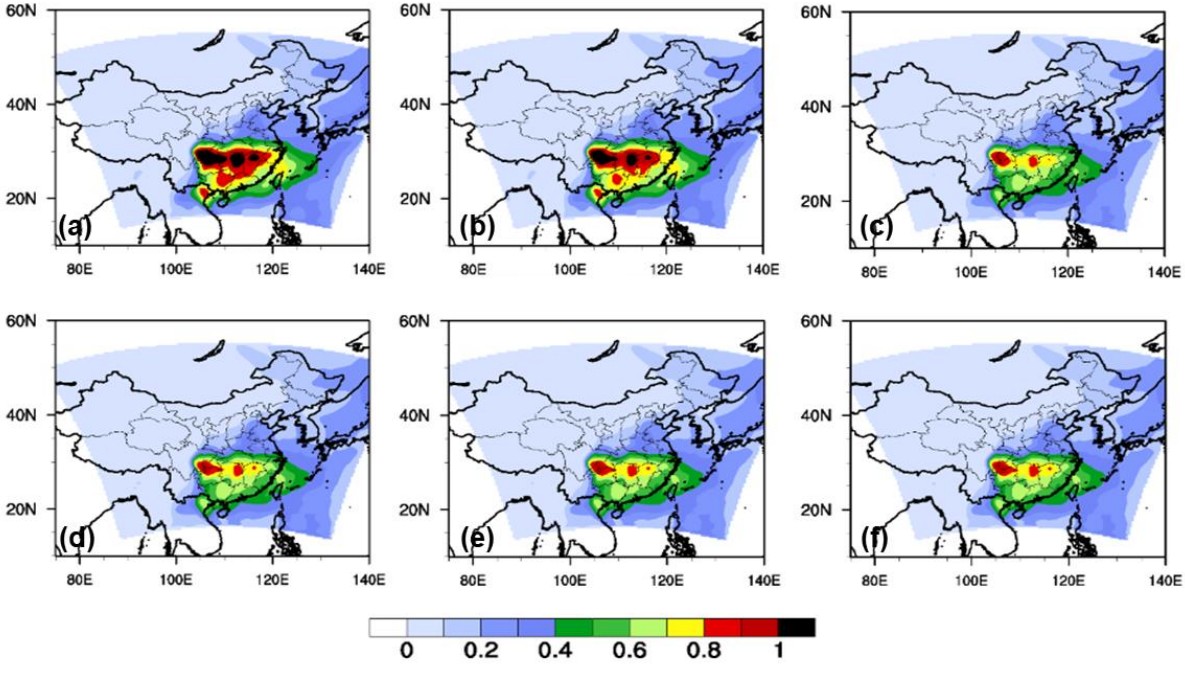

Figure 7. Monthly (January 2010) mean RIEMS-Chem modeled AOD from different simulations: control run
(default simulation with internal mixing assumption) (a), external mixing assumption (b), internal mixing
assumption but without BC (c), internal mixing assumption but with doubled BC (d), without dust and sea-salt

(e), and reduced RH (f)

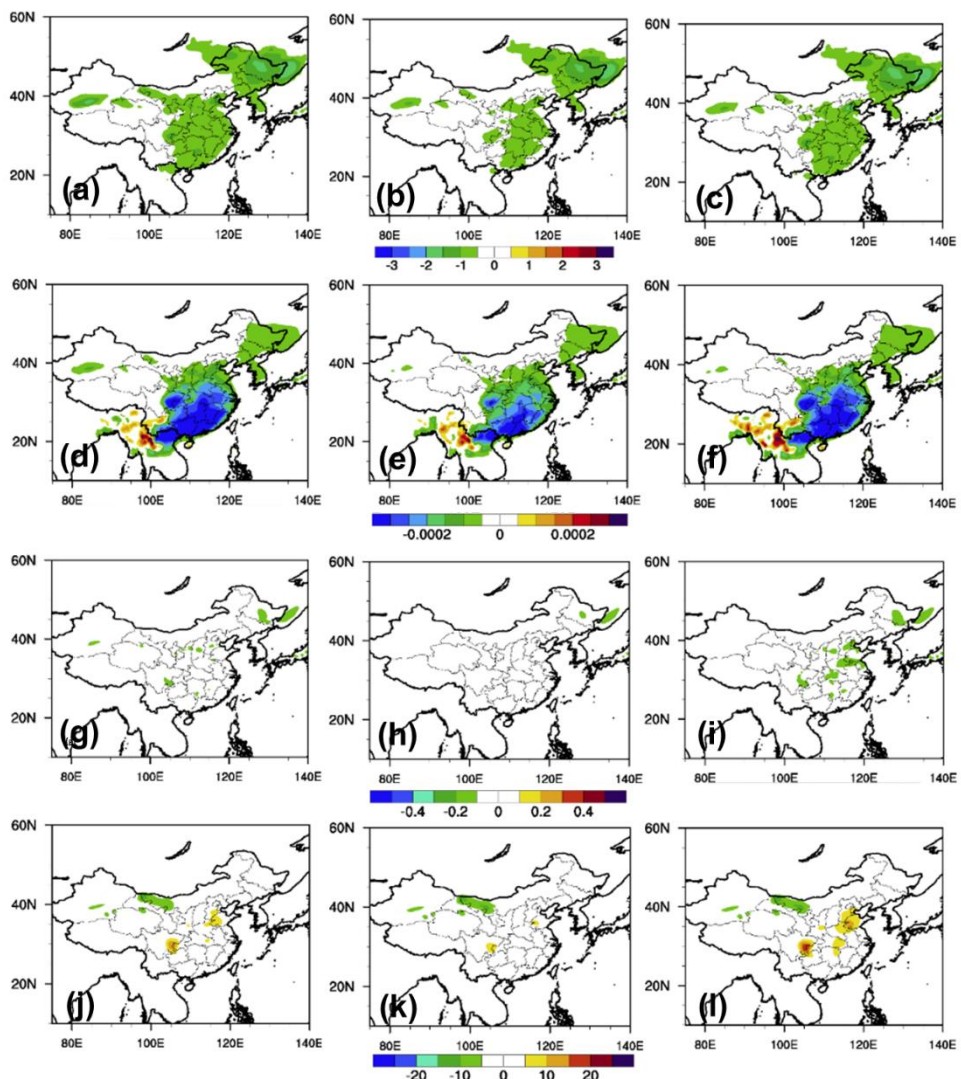


Figure 8. Monthly (January 2010) mean RIEMS-Chem modeled changes in T2 (ºC), Q2 (kg/kg), WS10 (m/s) and $PM_{2.5}$ (µg/m$^3$) from different simulations: external mixing assumption (first column), internal mixing assumption but without BC (second column) and internal mixing assumption but with doubled BC (third column)

454
455

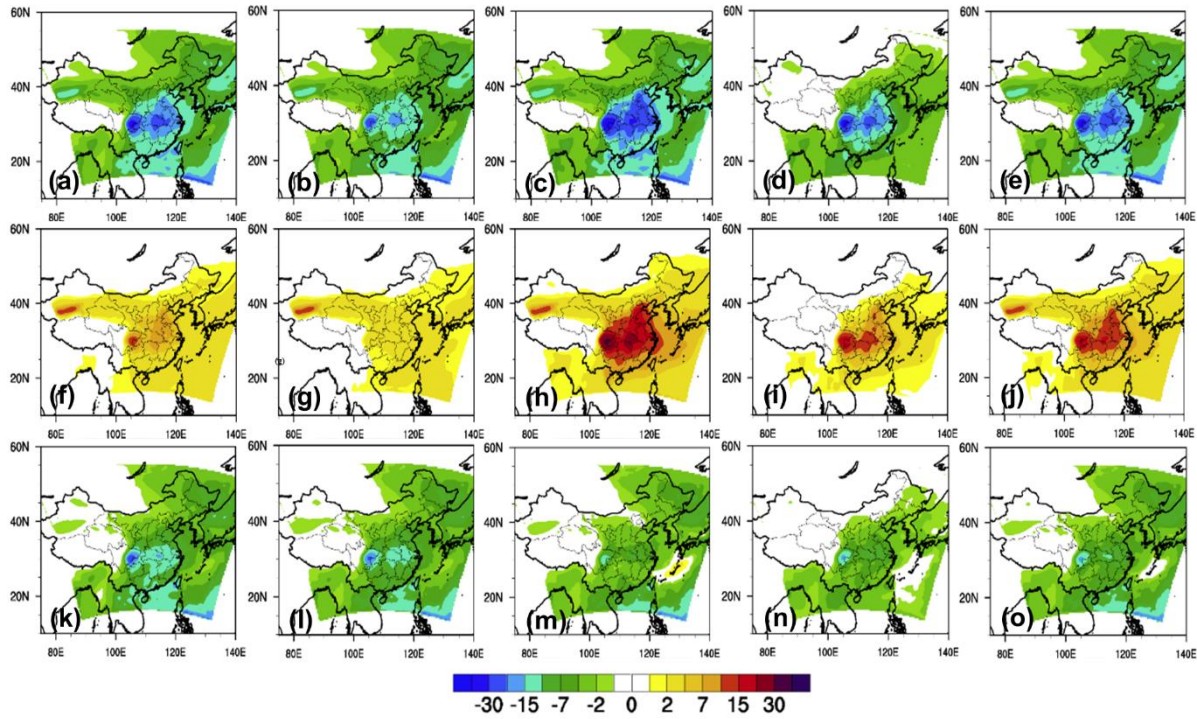

Figure 9. Monthly (January 2010) mean RIEMS-Chem modeled aerosol direct radiative forcing at the surface (a-e), inside the atmosphere (f-j) and at the top of the atmosphere (k-o) from different simulations: external mixing assumption (first column), internal mixing assumption but without BC (second column), internal mixing assumption but with doubled BC (third column), without dust and sea-salt (fourth column), and reduced RH (fifth column)

Table 1 Participating models in Topic 3

| Models | M1: WRF-Chem1 | M2: WRF-Chem2 | M3: NU-WRF1 | M4: NU-WRF2 | M5: RIEMS-Chem | M6: RegCCMS | M7: WRF-CMAQ |
|---|---|---|---|---|---|---|---|
| Modelling Group | Pusan National University | University of Iowa | USRA/NASA | USRA/NASA | Institute of Atmospheric Physics | Nanjing University | University of Tennessee |
| Grid Resolution | 45km | 50km | 45km | 15km | 60km | 50km | 45km |
| Vertical Layers | 40 layers to 50mb | 27 layers to 50mb | 60 layers to 20mb | 60 layers to 20mb | 16 layers to 100mb | 18 layers to 50mb | |
| Gas phase chemistry | RACM | CBMZ | RADM2 | RADM2 | CBM4 | CBM4 | SAPRC99 |
| Aerosols | MADE | MOSAIC-8bin | GOCART | GOCART | Sulfate, nitrate, ammonium, BC, OC, SOA, 5 bins of soil dust, and 5 bins of sea salt | Sulfate, nitrate, ammonium, BC and POC | AE06 |
| Chemical Boundary Conditions | Climatological data from NALROM | MOZART | MOZART GOCART | MOZART GOCART | GEOS-Chem | Climatological data | GEOS-Chem |

Table 2 Monthly Mean (January 2010) Aerosol Direct Radiative Forcing (W/m$^2$) and Changes in T2 (ºC), Q2 (g/kg), WS10 (0.1 m/s), and PM$_{2.5}$ (µg/m$^3$) for Beijing and Beijing-Tianjin-Hebei region (areas marked in Fig. S1)

| Beijing | M1 PNU | M2 UIOWA | M4 NASA | M5 IAP | M6 NJU | M7 UTK |
|---|---|---|---|---|---|---|
| ADRF TOA | -0.6 | -2.2 | -0.8 | -1.4 | -0.1 | -2.5 |
| ADRF ATM | 5.8 | 4.3 | 9.3 | 5.1 | 2.4 | 11.6 |
| ADRF SFC | -6.4 | -6.5 | -10.1 | -6.5 | -2.5 | -14.1 |
| T2 | -0.1 | -0.3 | -0.7 | -0.5 | -0.1 | 0.0 |
| Q2 | -1.2E-2 | -2.3E-2 | -6.4E-2 | -5.8E-2 | -5.8E-3 | 2.1E-2 |
| WS10 | -0.2 | -0.2 | -0.6 | -0.2 | 0.0 | -1.2 |
| PM$_{2.5}$ | 0.1 (0.2%) | 1.4 (1.6%) | 1.1 (1.7%) | 0.6 (1.4%) | -1.2 (-2.2%) | 1.0 (1.4%) |
| BTH | | | | | | |
| ADRF TOA | 0.2 | -1.4 | -0.3 | -2.6 | 0.0 | -2.4 |
| ADRF ATM | 7.3 | 5.4 | 10.1 | 6.3 | 3.6 | 14.6 |
| ADRF SFC | -7.1 | -6.8 | -10.4 | -8.9 | -3.6 | -17.0 |
| T2 | -0.2 | -0.4 | -0.8 | -0.6 | -0.2 | 0.0 |
| Q2 | -1.0E-2 | -2.5E-2 | -8.1E-2 | -7.6E-2 | -2.9E-2 | 2.5E-2 |
| WS10 | -0.2 | -0.2 | -0.9 | -0.4 | 0.1 | -0.9 |
| PM$_{2.5}$ | 0.8 (1.4%) | 1.8 (1.8%) | 2.2 (3.2 %) | 2.2 (3.9%) | -4.2 (-5.7%) | 2.2 (2.4%) |

Table 3 Monthly Mean (January 2010) Aerosol Direct Radiative Forcing and indirect Radiative Forcing ($W/m^2$) at the top of the atmosphere inferred from M4 and M5 (areas marked in Fig. S1)

| Beijing | direct | Indirect |
|---|---|---|
| M4 | -0.77 | -0.15 |
| M5 | -1.43 | -0.01 |
| BTH | | |
| M4 | -0.28 | 0.1 |
| M5 | -2.63 | -0.04 |

Table 4 Mean Aerosol (January 2010) Direct Radiative Forcing (W/m$^2$) and Changes in T2
(ºC), Q2 (g/kg), WS10 (0.1 m/s), and PM$_{2.5}$ (μg/m$^3$) for Beijing and Beijing-Tianjin-Hebei
(BTH) region averaged over January 17-19 2010 (areas marked in Fig. S1)

| Beijing | M1 PNU | M2 UIOWA | M4 NASA | M5 IAP | M6 NJU | M7 UTK |
|---|---|---|---|---|---|---|
| ADRF TOA | 2.6 | -1.4 | 1.8 | -3.0 | -0.6 | -3.3 |
| ADRF ATM | 18.6 | 9.8 | 21.5 | 13.3 | 7.3 | 32.3 |
| ADRF SFC | -16.0 | -11.2 | -19.7 | -16.3 | -7.9 | -35.6 |
| T2 | -0.5 | -0.5 | -1.7 | -1.3 | -0.1 | -1.5 |
| Q2 | -7.4E-2 | -6.2E-2 | -2.6E-1 | -1.8E-1 | -1.3E-2 | -9.2E-2 |
| WS10 | -0.1 | 0.2 | -2.3 | 0.4 | 0.5 | -0.8 |
| PM$_{2.5}$ | -1.1 (-0.9%) | 3.8 (1.7%) | 6.3 (3.8%) | 1.0 (0.8%) | -7.9 (-4.7%) | 1.3 (1.1%) |
| BTH | | | | | | |
| ADRF TOA | 1.4 | 0.1 | 4.9 | -4.6 | -0.7 | -3.8 |
| ADRF ATM | 18.3 | 12.0 | 19.1 | 13.2 | 10.0 | 36.1 |
| ADRF SFC | -16.9 | -11.9 | -14.2 | -17.8 | -10.7 | -39.9 |
| T2 | -0.6 | -0.7 | -1.6 | -1.2 | -0.3 | -1.5 |
| Q2 | -7.1E-2 | -8.2E-2 | -2.9E-1 | -2.0E-1 | -1.2E-1 | -8.9E-2 |
| WS10 | -0.3 | -0.4 | -2.5 | 0.0 | 0.3 | -0.9 |
| PM$_{2.5}$ | 2.9 (2.3%) | 8.5 (3.7%) | 5.3 (3.9%) | 5.3 (3.9%) | -10.5 (-6.2%) | 5.1 (2.7%) |
| Daytime PM$_{2.5}$ | | | | | | |
| Beijing | 2.4 (2.0%) | 8.5 (3.9%) | 8.4 (5.5%) | -0.7 (-0.6%) | -4.2 (-3.2%) | 10.7 (8.3%) |
| BTH | 6.0 (4.9%) | 12.9 (5.9%) | 6.6 (5.2%) | 5.3 (4.0%) | -6.2 (-3.8%) | 6.4 (3.8%) |
| | Up to 26.4 | Up to 55.4 | Up to 26.5 | Up to 21.1 | Up to 22.8 | Up to 60.9 |

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
