# Peer review of "Air Quality and Climate Change, Topic of the Model Inter-Comparison"

_Atmospheric Chemistry and Physics, 2019_

## Referee Comment (RC1) · Anonymous Referee #1 · 20 Sep 2019

General comments: The manuscript entitled "Air Quality and Climate Change, Topic 3 of the Model Inter-Comparison Study for Asia Phase III (MICS-Asia III), Part II: aerosol radiative effects and aerosol feedbacks discussed the estimates of aerosol radiative forcing, aerosol feedbacks and the dominant roles of direct forcing. The possible causes for the differences among the models were also analyzed by sensitivity simulation. Some interesting results have been found. However, the manuscript needs to be improved in writing and logically organization in its structure. I recommend to publish it after major revision.

[Figure]

Specific comments: 1. There are a few grammatical errors, please find a native speaker to proofread the paper. 2. The sections of manuscript need to be organized more logically in structure. 3. Lines 61-65: Some important previous work in China from the observational perspective have been ignored, including Huang et al., Satellite-based assessment of possible dust aerosols semi-direct effect on cloud water path over East Asia, 2006; Liu et al., Aerosol optical properties and radiative effect determined from sky-radiometer over Loess Plateau of Northwest China, 2011; Also, Line 82-89: some modeling work have been ignored, including Chen et al., Modeling the transport and radiative forcing of Taklimakan dust over the Tibetan Plateau, 2013; Chen et al., Dust modeling over East Asia during the summer of 2010 using the WRF-Chem model, 2018; Liu et al., Modeling study on the transport of summer dust and anthropogenic aerosols over the Tibetan Plateau, 2015; Jia et al., Estimation of the aerosol radiative effect over the Tibetan Plateau based on the latest CALIPSO product, 2018. These researches are highly relevant to the topic investigated here. 4. Line 66-69, please cite the previous researches as an illustration basis. 5. Lines 80-81, the meaning of theory and practice of studying aerosol feedbacks over Asia should be illustrated in detail. 6. Please add a section to describe the model and research methodology. Move the illustration in Line 90-101 and Line 157-159 to the new section. 7. Please give the detailed description when the abbreviation first appears (for example, M1, M2, M3...M7). 8. Please use the box or symbol to show the BTH region, Huabei province and Beijing in Figure 1. 9. Line 118-119, the description is inconsistent with Table 1. 10. Line 145-149, why do you only use M4 and M5 to provide direct and indirect aerosol radiative forcing? Why do you only use M5 to study the effects of aerosols mixing state, hygroscopic growth, black carbon and mineral dust. Obviously, the values given by these models are very different. Which should be illustrated. 11. Line 195-196, the description is inconsistent with Table 3. M2 (12.9) 12. Please add a discussion about the simulation performance of different models. 13. Line 271-272ïijŇplease give some evidence for that the effect of BC indicates smaller than that of other scattering aerosols. How about the effect of sulfate aerosol?

Technical corrections: 1. Line 110, 'In Sect. 2'??? Please check it. 2. Line 118, Please give the full spelling of BTH when the abbreviation appears at the first time. 3. Line 118, 'reports' should be changed to 'report'. 4. Line 121, Please give the full spelling of AOD. 5. Line 257, 'by dust' should be changed to 'of dust'.
* * *

---

## Referee Comment (RC2) · Anonymous Referee #2 · 18 Oct 2019

**General comments**

The paper is the second part of two papers discussing the results of the MICS-Asia III model inter-comparison exercise with special focus on the performance of online coupled air quality models in simulating high aerosol pollution in the North China Plain region during wintertime haze events. While the focus of first part is on the description of the design of the modelling exercise and the overall model performance, this paper focuses on the role of aerosol radiative forcing and aerosol meteorology interactions for six different models. By means of case studies with one of the models, the authors

investigate the sensitivity of aerosol radiative forcing to different aerosol descriptions. The current paper includes some interesting results and is generally worth to be published. However, some aspects need to discussed in more detail and the presentation quality must be improved for major parts of the paper. Therefore, I recommend to publish the paper after major revisions.

**Specific comments**

Although the paper is the second of two associated papers, it is necessary to add a section that gives a brief overview of the experimental design and model setup as well as the applied models. Furthermore, the name of the models should be connected to the abbreviations M1, M2, . . . This information is given in Part 1, so this is obviously not a secret. It could be looked up there, but including this information also in this paper (e.g. in Table 1) would enhance the paper's readability considerably. Please add also some information about the length of the simulated episode and the simulation setup. Was the entire episode covered by one single simulation or was the episode simulated as a sequence of shorter time slices? The way how the simulation is performed can affect the development of semi-direct effects to a certain amount.

Why is model M3 not included?

According to part 1 (Gao et al., 1918a) the simulation with WRF-CMAQ (M7) was performed with aerosol–radiation interactions turned off. If this is also the case here, this should be mentioned and eventual implications on the results should be discussed.

Line 78-80: Since e.g. Grell et al., 2011 (doi:10.5194/acp-11-5289-2011) and Yang et al. 2012 (https://doi.org/10.5194/acp-12-3045-2012) describe the development and implementation of aerosol-meteorology interactions into WRF-Chem, these papers should also be mentioned here and not only application papers. This holds of course also for the other models.

Lines 122-123: Please give a reference here.

[Figure]

Lines 125-126: Please try to explain this behavior.

Lines 133-134: What is 'other model treatments'?

Line 156-159 and caption of Fig. 3: The name of the model would be more helpful here.

Line 181: Why are the results for M6 so different?

Lines 228-242: These results should be discussed in relation to the results by Curci et al., 2015 (http://dx.doi.org/10.1016/j.atmosenv.2014.09.009) and similar studies.

Lines 253-254: Model names in addition to M5 and M5 would be helpful. Does M1 (which is also WRF-Chem) definitely not include any soil dust? It is possible to use MADE-Sorgam in combination with a dust option. Please clarify this in the paper.

Lines 285 (Language quality): 'Previous paper': Which previous paper?

Line 291: What is the reason for this?

Line 299: Which studies?

Caption of Fig. 2: Which month?

Caption of Table 1 and 2: To which area do the results given in the tables refer to?

The language quality must be improved by consulting a native speaker or a language editing service.

**Minor points**

Line 101, 175, 188: Start a new paragraph here (and everywhere else, where you start to discuss a new topic).

Lines 104-105, 122, 133 (could be resulted), 216-220, 291, 303-304 and many other lines: Odd language

Line 122: 'an external assumption': About what?

Line 122: Why 'also'? Which other models use also the assumption of external mixing?

Line 169: A word seems to missing here.

Line 225: A reference should be given here.

Line 282: A reference should be given here.

References in Figure captions: 2018a or 2018b? Better mention the models instead of the institutions (eventually not necessary for all figures)

Line 450: Use either complete list of authors (Forkel, R., Balzarini, A., Baró, R., Bianconi, R., Curci, G., Jiménez-Guerrero, P., Hirtl, M., Honzak, L., Lorenz, C., Im, U., Pérez, J. L., **Pirovano, G., José, R. S., Tuccella, P., Werhahn, J., and Zabkar, R.**: Analysis of the WRF-Chem contributions to AQMEII phase2 with respect to aerosol radiative feedbacks on meteorology and pollutant distributions, Atmos. Environ., 115, 630–645, 2015.) or 'et al' after the third author.

---

## Author Comment (AC1) · 3 Dec 2019

General comments: The manuscript entitled "Air Quality and Climate Change, Topic 3 of the Model Inter-Comparison Study for Asia Phase III (MICS-Asia III), Part II: aerosol radiative effects and aerosol feedbacks discussed the estimates of aerosol radiative forcing, aerosol feedbacks and the dominant roles of direct forcing. The possible causes for the differences among the models were also analyzed by sensitivity simulation. Some interesting results have been found. However, the manuscript needs to be improved in writing and logically organization in its structure. I recommend publishing it after major revision.

➢ Reply: Thanks for the valuable comments. We have revised the manuscript following your comments, which are shown below.

Specific comments:

1. There are a few grammatical errors, please find a native speaker to proofread the paper.

➢ Reply: We, including a native speaker, have carefully read the manuscript and edited to avoid grammatical errors.

2. The sections of manuscript need to be organized more logically in structure.

➢ Reply: The manuscript discusses aerosol radiative forcing as it is closely connected with aerosol feedbacks, so we put it in Sect. 3 while aerosol feedbacks in Sect. 4. In Sect. 5, simulations were conducted to figure out the reasons for differences. We added Sect. 2 to describe the design of the experiment to make it easier to read.

3. Lines 61-65: Some important previous work in China from the observational perspective have been ignored, including

Huang et al., Satellite-based assessment of possible dust aerosols semi-direct effect on cloud water path over East Asia, 2006;

Liu et al., Aerosol optical properties and radiative effect determined from sky-radiometer over Loess Plateau of Northwest China, 2011;

Also, Line 82-89: some modeling work have been ignored, including

Chen et al., Modeling the transport and radiative forcing of Taklimakan dust over the Tibetan Plateau, 2013;

Chen et al., Dust modeling over East Asia during the summer of 2010 using the WRF-Chem model, 2018;

Liu et al., Modeling study on the transport of summer dust and anthropogenic aerosols over the Tibetan Plateau, 2015;

Jia et al., Estimation of the aerosol radiative effect over the Tibetan Plateau based on the latest CALIPSO product, 2018.

These researches are highly relevant to the topic investigated here.

> ➢ Reply: Thanks for mentioning these important studies. We have added these references in the revised manuscript.
> ➢ Huang, J., Lin, B., Minnis, P., Wang, T., Wang, X., Hu, Y., Yi, Y. and Ayers, J.K.: Satellite-based assessment of possible dust aerosols semi-direct effect on cloud water path over East Asia, Geophy. Res. Let., 33(19), https://doi.org/10.1029/2006GL026561, 2006.
> ➢ Liu, Y., Huang, J., Shi, G., Takamura, T., Khatri, P., Bi, J., Shi, J., Wang, T., Wang, X. and Zhang, B.: Aerosol optical properties and radiative effect determined from sky-radiometer over Loess Plateau of Northwest China, Atmos. Chem. Phys., 11(22), pp.11455-11463, https://doi.org/10.5194/acp-11-11455-2011, 2011.
> ➢ Chen, S., Huang, J., Zhao, C., Qian, Y., Leung, L.R. and Yang, B.: Modeling the transport and radiative forcing of Taklimakan dust over the Tibetan Plateau: A case study in the summer of 2006, Jour. Geophy. Res.: Atmos., 118(2), pp.797-812, https://doi.org/10.1002/jgrd.50122, 2013.
> ➢ Chen, S., Yuan, T., Zhang, X., Zhang, G., Feng, T., Zhao, D., Zang, Z., Liao, S., Ma, X., Jiang, N. and Zhang, J.: Dust modeling over East Asia during the summer of 2010 using the WRF-Chem model, Jour. Quan. Spec. Rad. Tran., 213, pp.1-12, https://doi.org/10.1016/j.jqsrt.2018.04.013, 2018.
> ➢ Liu, Y., Sato, Y., Jia, R., Xie, Y., Huang, J. and Nakajima, T.: Modeling study on the transport of summer dust and anthropogenic aerosols over the Tibetan Plateau, Atmos. Chem. Phys., 15(21), pp.12581-12594, https://doi.org/10.5194/acp-15-12581-2015, 2015.
> ➢ Jia, R., Liu, Y., Hua, S., Zhu, Q. and Shao, T.: Estimation of the aerosol radiative effect over the Tibetan Plateau based on the latest CALIPSO product, Jour. Met. Res., 32(5), pp.707-722, https://doi.org/10.1007/s13351-018-8060-3, 2018.

4. Line 66-69, please cite the previous researches as an illustration basis.

> ➢ Reply: We have included these studies in the revised manuscript:
> ➢ "Aerosols change weather and climate via the following pathways: they absorb and scatter solar and thermal radiation to alter the radiative balance of the earth-atmosphere system (Liu et al., 2011; Jia et al., 2018)";
> ➢ "The suppression of cloud convection induced by direct effects of absorbing aerosols is called the semi-direct effect (Huang et al., 2006; Lohmann and Feichter, 2005)."
> ➢ "including WRF-Chem (the Weather Research Forecasting model coupled with Chemistry, Chen et al., 2013, 2018; Gao et al., 2016, 2017; Liu et al., 2015),"

5. Lines 80-81, the meaning of theory and practice of studying aerosol feedbacks over Asia should be illustrated in detail.

- ➢ Reply: We have added descriptions of the theory in the revised manuscript: "High concentrations of aerosols would enhance the stability of boundary layer due to reductions in radiation that reach the surface, which in turn cause further increases in PM2.5 concentrations (Ding et al., 2016; Gao et al., 2016)".

- ➢ More explanations of related studies can be found in Sect. 3: "These results can be compared to previous studies. The contributions of aerosol-radiation feedback to haze formation in China have been investigated in many previous studies (Ding et al., 2016; Gao et al., 2015; Gao et al., 2016; Liu et al., 2018; J. Wang et al., 2014; Z. Wang et al., 2014; Wang et al., 2015; Wu et al., 2019; Zhang et al., 2015; Zhang et al., 2018; Zhong et al., 2018), but the reported values partly diverge."

6. Please add a section to describe the model and research methodology. Move the illustration in Line 90-101 and Line 157-159 to the new section.

- ➢ Reply: Model descriptions, research methodology and model evaluations were provided in a companion paper, part I:

- ➢ Gao, M., Han, Z., Liu, Z., Li, M., Xin, J., Tao, Z., Li, J., Kang, J.-E., Huang, K., Dong, X., Zhuang, B., Li, S., Ge, B., Wu, Q., Cheng, Y., Wang, Y., Lee, H.-J., Kim, C.-H., Fu, J. S., Wang, T., Chin, M., Woo, J.-H., Zhang, Q., Wang, Z., and Carmichael, G. R.: Air quality and climate change, Topic 3 of the Model Inter-Comparison Study for Asia Phase III (MICS-Asia III) – Part 1: Overview and model evaluation, Atmos. Chem. Phys., 18, 4859–4884, https://doi.org/10.5194/acp-18-4859-2018, 2018.

- ➢ In the revised manuscript, we have added a section to describe the activity.

- ➢ "2 Overview of MICS-Asia III Topic 3 The participants were requested to use common emissions to simulate air quality during January 2010 and submit requested model variables. Table 1 summarizes the characteristics of the participating models. These models include one application of the Weather Research Forecasting model coupled with Chemistry (WRF-Chem; Fast et al., 2006; *Grell et al., 2005*) by Pusan National University (PNU) (M1); one application of the WRF-Chem model by the University of Iowa (UIOWA) (M2); two applications (two domains: 45 and 15 km horizontal resolutions) of the National Aeronautics and Space Administration (NASA) Unified WRF (NU-WRF; *Peters-Lidard*

*et al., 2015*) model by the Universities Space Research Association (USRA) and NASA's Goddard Space Flight Center (M3 and M4); one application of the Regional Integrated Environment Modeling System with Chemistry (RIEMS-Chem; *Han et al., 2010*) by the Institute of Atmospheric Physics (IAP), Chinese Academy of Sciences (M5); one application of the coupled Regional Climate Chemistry Modeling System (RegCCMS; *Wang et al., 2010*) from Nanjing University (M6); and one application of the coupled WRF-CMAQ (Community Multiscale Air Quality) model by the University of Tennessee at Knoxville (UTK) (M7). A new Asian emission inventory was developed for MICS-III by integrating state-of-the-art national or regional inventories to support this model intercomparison study (*Li et al., 2017*), which was provided to all modeling groups, along with biogenic emissions, biomass burning emissions, emissions from air and ship transport, volcano emissions, and dust emissions. Simulations of two global chemical transport models (e.g., GEOS-Chem (The Goddard Earth Observing System Model-Chemistry) and MOZART (Model for OZone And Related chemical Tracers)) were used as boundary conditions for MICS-Asia III. Comprehensive model evaluations suggest that all models could capture the observed near-surface temperature and water vapor mixing ratio, but overestimated near-surface wind speeds to varying degrees. Participating models were able to represent the observed daily maximum downward shortwave radiation, particularly low values during haze days, and the observed variations of air pollutants, including $SO_2$, $NO_x$, CO, $O_3$, $PM_{2.5}$, and $PM_{10}$. However, large differences in the models were found in the predicted $PM_{2.5}$ chemical compositions."

7. Please give the detailed description when the abbreviation first appears (for example, M1, M2, M3. . .M7).

   ➢ Reply: M1, M2…M7 represent different participating models, which were documented in our paper part I. To make it easier to read, we added Table 1 into the revised manuscript: "from M1 (a), M2 (b), M4 (c), M5 (d), M6 (e), M7 (f) (Table 1:".

8. Please use the box or symbol to show the BTH region, Huabei province and Beijing in Figure 1.

   ➢ Reply: BTH and Beijing are small areas in Figure 1 and it is not clear if we mark them in Figure 1. Thus, we add Figure S1 in the supplement to display the BTH region (marked with blue) and Beijing (shown using the green arrow). In the revised manuscript, we change Huabei province to the BTH region.

[Figure]

Figure S1. The Beijing-Tianjin-Hebei region is marked with blue and Beijing is shown with green arrow.

9. Line 118-119, the description is inconsistent with Table 1.

➢ Reply: We have changed it to make it consistent.

10. Line 145-149, why do you only use M4 and M5 to provide direct and indirect aerosol radiative forcing? Why do you only use M5 to study the effects of aerosols mixing state, hygroscopic growth, black carbon and mineral dust. Obviously, the values given by these models are very different. Which should be illustrated.

➢ Reply: MICS-Asia is a volunteer-based model inter-comparison activity. Only limited model outputs were requested. Most modeling groups did conduct extra numerical experiments to distinguish direct and indirect forcing. Thus, we only include the discussions based on M4 and M5, and the results from these two model applications are consistent that direct forcing dominates the total forcing.

➢ Please notice that we have updated the results from M5 in the revised manuscript as some mistakes were found for M5 in the calculations of optical properties. The updated results show higher agreements with other models, but M5 still produces the largest (negative)

radiative forcing at TOA and the second largest forcing at the surface in both the BTH region and Beijing.

> It is of great importance to understand the drivers for the differences, and the IAP group (M5) volunteered to conduct additional simulations. We agree that different models would behave differently in the sensitivity simulations. To address this, we added comparison and discussion in the revised manuscript to provide a clearer picture of how these factors would influence the results.

> "Large uncertainties still remain in the estimates of the role of BC in aerosol feedbacks relative to scattering aerosols. Gao et al. (2016) suggested that the impacts of BC on boundary layer height and PM2.5 concentrations can account for as high as 60% of the total aerosol feedbacks in the North China Plain at 2 p.m., although it only accounts for a small share of PM in terms of mass concentration. Qiu et al. (2017) indicated that PM2.5 concentrations averaged over the North China Plain increased by 16.8% and 1.0% due to scattering aerosols and BC, respectively. It should be noted that most participating models, including RIEMS-Chem, tend to underpredict the total mass concentrations of scattering aerosols (inorganic and organic aerosols) by up to a factor of two over the study period, leading to overestimation of the contribution of BC."

11. Line 195- 196, the description is inconsistent with Table 3. M2 (12.9)

> Reply: We have changed the upper range to 12.9.

12. Please add a discussion about the simulation performance of different models.

> Reply: We evaluated all models in the companion paper part I: "model evaluations have been archived in Gao et al. (2018a)."

> We did include discussions in this manuscript:

> "Model evaluation of PM2.5 composition in Gao et al. (2018a) reveals that M4 overpredicts strong scattering organic carbon, which could be one of the reasons for higher temperature reduction."

> "As suggested in model evaluation, sulfate and organic aerosol concentrations are generally underestimated by most models in this study, except that M4 overestimate organic aerosol (Gao et al., 2018a). These were attributed to the missing multiphase

oxidation mechanisms of SO2, and different secondary organic aerosol (SOA) formation mechanisms in these models (Gao et al., 2018a)."

➢ Following your suggestion, we added more discussions of model evaluation in the revised manuscript: "Comprehensive model evaluations suggest that all models could capture the observed near-surface temperature and water vapor mixing ratio, but overestimated near-surface wind speeds to varying degrees. Participating models were able to represent the observed daily maximum downward shortwave radiation, particularly low values during haze days, and the observed variations of air pollutants, including SO2, NOx, CO, O3, PM2.5, and PM10. However, large differences in the models were found in the predicted PM2.5 chemical compositions. "

13. Line 271-272 please give some evidence for that the effect of BC indicates smaller than that of other scattering aerosols. How about the effect of sulfate aerosol?

➢ Reply: To avoid confusion, we rewrote the sentence in the revised manuscript and added discussions:

➢ "Two sets of simulations, namely without BC and with doubled BC concentrations, were conducted to examine the influences of BC on aerosol radiative forcing and feedbacks. In the control case, the aerosol induced changes in monthly mean surface air temperature, wind speed and PM2.5 values are -0.47 °C, -0.03 m/s and 1.5 $\mu$g/m3 for the BTH region, respectively. When BC is not included (only scattering aerosols and dust), the corresponding aerosol induced changes are -0.37 °C, -0.02 m/s and 0.7 $\mu$g/m3, respectively. When BC concentrations are doubled, these values change to -0.52 °C, -0.04 m/s and 2.2 $\mu$g/m3, respectively. The comparison between the control case and two additional sensitivity cases indicates that the changes caused by BC are comparable to those by scattering aerosols. The contribution of BC to aerosol feedbacks can reach up to 40~50%. It is also found that the influence of BC on aerosol feedbacks with internal mixing assumption is larger than that with external mixing assumption (Figure not shown).

➢ Large uncertainties still remain in the estimates of the role of BC in aerosol feedbacks relative to scattering aerosols. Gao et al. (2016) suggested that the impacts of BC on boundary layer height and PM2.5 concentrations can account for as high as 60% of the total aerosol feedbacks in the North China Plain at 2 p.m., although it only accounts for a small share of PM in terms of mass concentration. Qiu et al. (2017) indicated that PM2.5

concentrations averaged over the North China Plain increased by 16.8% and 1.0% due to scattering aerosols and BC, respectively. It should be noted that most participating models, including RIEMS-Chem, tend to underpredict the total mass concentrations of scattering aerosols (inorganic and organic aerosols) by up to a factor of two over the study period, leading to overestimation of the contribution of BC."

➢ In the companion paper part I, simulations of aerosol components were validated against observations in Beijing. Observations show that concentrations of sulfate (daily mean exceeding 60 μg/m3) are comparable to nitrate, and are about 30% lower than the concentrations of OC during haze days. The hygroscopicity of sulfate and nitrate is larger than that of OC. Most of the models underpredict concentrations of total scattering aerosols (sulfate, nitrate and OC) by up to a factor of two, while models show reasonably good skills for BC. These results imply that the effect of scattering aerosols on aerosol feedbacks might have been underestimated.

➢ There were no simulations in MICS-Asia to separate the effect of sulfate. In the revised manuscript, we added sentences to acknowledge this limitation: "Huang et al. (2015) separated the contributions of different aerosol components to aerosol direct radiative forcing, highlighting the roles of BC and sulfate. Future studies are also needed to separate the effects of other aerosol components, including sulfate, on aerosol feedbacks. "

➢ Qiu, Y., Liao, H., Zhang, R. and Hu, J.: Simulated impacts of direct radiative effects of scattering and absorbing aerosols on surface layer aerosol concentrations in China during a heavily polluted event in February 2014, Jour. Geophy. Res.: Atmos., 122(11), pp.5955-5975, https://doi.org/10.1002/2016JD026309, 2017.

➢ Huang, X., Song, Y., Zhao, C., Cai, X., Zhang, H. and Zhu, T.: Direct radiative effect by multicomponent aerosol over China, Jour. Clim., 28(9), pp.3472-3495, https://doi.org/10.1175/JCLI-D-14-00365.1, 2015.

Technical corrections:

1. Line 110, 'In Sect. 2'??? Please check it.

➢ Reply: We've changed to Sect. 3.

2. Line 118, Please give the full spelling of BTH when the abbreviation appears at the first time.

> ➤ Reply: We've changed accordingly.

3. Line 118, 'reports' should be changed to 'report'.

> ➤ Reply: We've changed accordingly.

4. Line 121, Please give the full spelling of AOD.

> ➤ Reply: We've changed accordingly.

5. Line 257, 'by dust' should be changed to 'of dust'.

> ➤ Reply: We've changed accordingly.

**Anonymous Referee #2**

General comments

The paper is the second part of two papers discussing the results of the MICS-Asia III model inter-comparison exercise with special focus on the performance of online coupled air quality models in simulating high aerosol pollution in the North China Plain region during wintertime haze events. While the focus of first part is on the description of the design of the modelling exercise and the overall model performance, this paper focuses on the role of aerosol radiative forcing and aerosol meteorology interactions for six different models. By means of case studies with one of the models, the authors investigate the sensitivity of aerosol radiative forcing to different aerosol descriptions. The current paper includes some interesting results and is generally worth to be published. However, some aspects need to be discussed in more detail and the presentation quality must be improved for major parts of the paper. Therefore, I recommend publishing the paper after major revisions.

> ➤ Reply: Thanks for the valuable comments. We have revised the manuscript following your comments, especially the presentation quality. Detailed modifications are shown below.

Specific comments

Although the paper is the second of two associated papers, it is necessary to add a section that gives a brief overview of the experimental design and model setup as well as the applied models. Furthermore, the name of the models should be connected to the abbreviations M1, M2, . . . This information is given in Part 1, so this is obviously not a secret. It could be looked up there, but including this information also in this paper (e.g. in Table 1) would enhance the paper's readability considerably.

- ➢ Reply: Thanks for the good suggestion. In the revised manuscript, we have added a section to describe the activity.
- ➢ We have added Table 1 to connect model names and abbreviations in the revised manuscript.
- ➢ "2 Overview of MICS-Asia III Topic 3 The participants were requested to use common emissions to simulate air quality during January 2010 and submit requested model variables. Table 1 summarizes the characteristics of the participating models. These models include one application of the Weather Research Forecasting model coupled with Chemistry (WRF-Chem; Fast et al., 2006; *Grell et al., 2005*) by Pusan National University (PNU) (M1); one application of the WRF-Chem model by the University of Iowa (UIOWA) (M2); two applications (two domains: 45 and 15 km horizontal resolutions) of the National Aeronautics and Space Administration (NASA) Unified WRF (NU-WRF; *Peters-Lidard et al., 2015*) model by the Universities Space Research Association (USRA) and NASA's Goddard Space Flight Center (M3 and M4); one application of the Regional Integrated Environment Modeling System with Chemistry (RIEMS-Chem; *Han et al., 2010*) by the Institute of Atmospheric Physics (IAP), Chinese Academy of Sciences (M5); one application of the coupled Regional Climate Chemistry Modeling System (RegCCMS; *Wang et al., 2010*) from Nanjing University (M6); and one application of the coupled WRF-CMAQ (Community Multiscale Air Quality) model by the University of Tennessee at Knoxville (UTK) (M7). A new Asian emission inventory was developed for MICS-III by integrating state-of-the-art national or regional inventories to support this model intercomparison study (*Li et al., 2017*), which was provided to all modeling groups, along with biogenic emissions, biomass burning emissions, emissions from air and ship transport, volcano emissions, and dust emissions. Simulations of two global chemical transport models (e.g., GEOS-Chem (The Goddard Earth Observing System Model-Chemistry) and

MOZART (Model for OZone And Related chemical Tracers)) were used as boundary conditions for MICS-Asia III. Comprehensive model evaluations suggest that all models could capture the observed near-surface temperature and water vapor mixing ratio, but overestimated near-surface wind speeds to varying degrees. Participating models were able to represent the observed daily maximum downward shortwave radiation, particularly low values during haze days, and the observed variations of air pollutants, including $SO_2$, $NO_x$, CO, $O_3$, $PM_{2.5}$, and $PM_{10}$. However, large differences in the models were found in the predicted $PM_{2.5}$ chemical compositions."

Please add also some information about the length of the simulated episode and the simulation setup. Was the entire episode covered by one single simulation or was the episode simulated as a sequence of shorter time slices? The way how the simulation is performed can affect the development of semi-direct effects to a certain amount.

➤ Reply: We have added one sentence to describe this: "The entire month of January 2010 was simulated and covered by one single simulation for each participating model."

Why is model M3 not included? According to part 1 (Gao et al., 2018a) the simulation with WRF-CMAQ (M7) was performed with aerosol–radiation interactions turned off. If this is also the case here, this should be mentioned and eventual implications on the results should be discussed.

➤ Reply: Nudging of meteorological variables were applied for M3, so the simulated feedbacks are not apparent. We decided not to include in the comparison. In model evaluation shown in Gao et al. (2018a), the results of M7 are from a simulation with aerosol-radiation interactions turned off, but the results from this study are based on online simulation of M7. As the current paper dicussses aerosol feedbacks, in which results must come from online simulations, it will not lead to confusion with whether aerosol-radiation interactions in M7 are on or off.

Line 78-80: Since e.g. Grell et al., 2011 (doi:10.5194/acp-11-5289-2011) and Yang et al. 2012 (https://doi.org/10.5194/acp-12-3045-2012) describe the development and implementation of aerosol-meteorology interactions into WRF-Chem, these papers should also be mentioned here and not only application papers. This holds of course also for the other models.

➤ Reply: We have added the suggested references in the revised manuscript.

➢ Grell, G. A., Peckham, S. E., Schmitz, R., McKeen, S. A., Frost, G., Skamarock, W. C., and Eder, B.: Fully coupled "online" chemistry within the WRF model, Atmos. Environ. 39, 6957–6975, 2005.

➢ Saide, P. E., Spak, S. N., Carmichael, G. R., Mena-Carrasco, M. A., Yang, Q., Howell, S., Leon, D. C., Snider, J. R., Bandy, A. R., Collett, J. L., Benedict, K. B., de Szoeke, S. P., Hawkins, L. N., Allen, G., Crawford, I., Crosier, J., and Springston, S. R.: Evaluating WRF-Chem aerosol indirect effects in Southeast Pacific marine stratocumulus during VOCALS-REx, Atmos. Chem. Phys., 12, 3045-3064, https://doi.org/10.5194/acp-12-3045-2012, 2012.

➢ Yang, Q., W. I. Gustafson Jr., Fast, J. D., Wang, H., Easter, R. C., Morrison, H., Lee, Y.-N., Chapman, E. G., Spak, S. N., and Mena-Carrasco, M. A.: Assessing regional scale predictions of aerosols, marine stratocumulus, and their interactions during VOCALS-REx using WRF-Chem, Atmos. Chem. Phys., 11, 11951–11975, doi:10.5194/acp-11-11951-2011, 2011.

Lines 122-123: Please give a reference here.

➢ Reply: We have added a reference in the revised manuscript: "M6 also use an external assumption which likely cause weaker absorption and ADRF in the atmosphere (Conant et al., 2003).".

➢ Conant, W.C., Seinfeld, J.H., Wang, J., Carmichael, G.R., Tang, Y., Uno, I., Flatau, P.J., Markowicz, K.M. and Quinn, P.K.: A model for the radiative forcing during ACE-Asia derived from CIRPAS Twin Otter and R/V Ronald H. Brown data and comparison with observations, Jour. Geophy. Res.: Atmos., 108(D23), https://doi.org/10.1029/2002JD003260, 2003.

Lines 125-126: Please try to explain this behavior.

➢ Reply: We added explanation in the revised manuscript: "This is related to the strong negative forcing at the surface and predicted high concentrations of sulfate by M5 (Gao et al., 2018a)."

Lines 133-134: What is 'other model treatments'?

➢ Reply: Other model treatments include parameterization of hygroscopicity, including dust or not, etc. To avoid confusion, we change the sentence to: "while discrepancies among models could be resulted from assumptions for mixing state and other model treatments (parameterization of hygroscopicity, mineral dust, etc.)."

Line 156-159 and caption of Fig. 3: The name of the model would be more helpful here.

➢ Reply: We have added names of models here and Table 1 to describe these models.

➢ (Table 1: M1: WRF-Chem, Pusan National University; M2: WRF-Chem, University of Iowa; M4: NU-WRF, NASA; M5: RIEMS-Chem, Institute of Atmospheric Physics; M6: RegCCMS, Nanjing University; M7: WRF-CMAQ, University of Tennessee; Gao et al., 2018a).

Line 181: Why are the results for M6 so different?

➢ Reply: Most other models use WRF as the meteorological model while M6 uses a very different climate model. In additional, lots of parameterization schemes used in M6 are different. For example, other models use a kappa parameterization to describe aerosol hygroscopic growth, while M6 uses a different hygroscopic growth scheme following Kiehl and Briegleb (1993). M6 produces way too high concentrations of sulfate in Beijing. All these factors make M6 very different.

Lines 228-242: These results should be discussed in relation to the results by Curci et al., 2015 (http://dx.doi.org/10.1016/j.atmosenv.2014.09.009) and similar studies.

➢ Reply: In the revised manuscript, we added comparison with Curci et al., 2015 and other similar studies:

➢ "In the control case, a homogeneous mixture of inorganic aerosols and BC is assumed. The refractive index of this mixture is estimated using the volume-weighted average of the refractive index of individual component. The size of the mixture is prescribed to be the maximum size of the mixed aerosol components. For example, the size of the mixture of

sulfate and BC is set to be equal to sulfate, assuming a small BC particle sticking to a larger sulfate particle. An additional simulation was conducted with the assumption of external mixing, and the corresponding results are displayed in Fig. 7-9. For external mixing assumption, each aerosol component is considered individually, and the total AOD is calculated as the sum of extinction by each aerosol component. Compared with internal mixing assumption, results from external mixing assumption generally exhibit a weaker (negative) ADRF at the surface (~15%), a stronger (negative) ADRF at TOA (~50%) and a decreased (positive) ADRF in the atmosphere (~30%) (Fig. 9a, 9f, 9k). These reponses of ADRF to aerosol mixing state inferred by this study are consistent with those from Conant et al. (2003). Curci et al. (2015) reported lower AOD with internal mixing assumption than external mixing assumption, because aerosol mass was distributed more to larger particles. As a result, fewer scattering agents are estimated, leading to lower AOD. These differences also suggest that the effects of mixing state on radiative forcing may differ under different treatments. With external mixing assumption, M5 predicts smaller aerosol feedbacks (changes in surface meteorological variables and PM2.5 concentrations, Fig. 8a, 8d, 8g, and 8j) than the estimates with internal mixing assumption. The monthly averaged changes in surface air temperature, wind speed and PM2.5 values are -0.47 °C, -0.03 m/s and 1.5 $\mu$g/m3 for the BTH region with internal mixing assumption, while the corresponding values change to -0.46 °C, -0.02 m/s and 1.2 $\mu$g/m3 with external mixing assumption. These differences emphasize the important influences of aerosol mixing state on ADRF and aerosol feedbacks. Aerosol mixing states can vary with time and location. Measurements in North China suggest that aerosols are partially internally mixed, and the fraction of internal mixing increased from clean to haze periods (Li et al., 2014)."

➢ Conant, W.C., Seinfeld, J.H., Wang, J., Carmichael, G.R., Tang, Y., Uno, I., Flatau, P.J., Markowicz, K.M. and Quinn, P.K.: A model for the radiative forcing during ACE-Asia derived from CIRPAS Twin Otter and R/V Ronald H. Brown data and comparison with observations, Jour. Geophy. Res.: Atmos., 108(D23), https://doi.org/10.1029/2002JD003260, 2003.

Lines 253-254: Model names in addition to M5 and M5 would be helpful. Does M1 (which is also WRF-Chem) definitely not include any soil dust? It is possible to use MADE-Sorgam in combination with a dust option. Please clarify this in the paper.

➢ Reply: All models have the options to include dust, but in some applications, the modelers did not turn this option on. We have changed the sentence to make it clearer: "M5 (RIEMS-Chem) includes all anthropogenic aerosols and dust, sea salt, while the other models except M2 (WRF-Chem, University of Iowa) do not consider natural dust in their model settings."

Lines 285 (Language quality): 'Previous paper': Which previous paper?

➢ Reply: We have changed "previous paper" to Gao et al. (2018a).

Line 291: What is the reason for this?

➢ Reply: ADRF at the TOA is the sum of ADRF at the surface (negative) and ADRF inside the atmosphere (positive). the ADRF at the TOA can be either positive and negative, depending on the relative magnitudes of ADRF at the surface and inside the atmosphere. Most of the model results show alternating sign of positive and negative values in the distribution of ADRF at the TOA, in contrast to the consistent negative and positive values of ADRF at the surface and in the atmosphere. This is related to the distribution of predicted relative importance of scattering and absorbing aerosols.

Line 299: Which studies?

➢ Reply: We have changed the sentence to "Our findings differ from previous studies (Ding et al., 2016; Gao et al., 2015; Gao et al., 2016; Liu et al., 2018; J. Wang et al., 2014; Z. Wang et al., 2014; Wang et al., 2015; Wu et al., 2019; Zhang et al., 2015; Zhang et al., 2018; Zhong et al., 2018)."

➢

Caption of Fig. 2: Which month?

➢ Reply: To avoid confusion, we have added "(January 2010)" into all captions of figures.

Caption of Table 1 and 2: To which area do the results given in the tables refer to?

➢ Reply: We add Figure S1 in the supplement to display the BTH region (marked with blue) and Beijing (shown using the green arrow). These regions are defined with political boundaries. We also add "(areas marked in Fig. S1)" in captions of Tables

[Figure]

Figure S1. The Beijing-Tianjin-Hebei region is marked with blue and Beijing is shown with green arrow.

The language quality must be improved by consulting a native speaker or a language editing service.

➢ Reply: We (including a native speaker) have carefully checked the language and improved the quality.

Minor points

Line 101, 175, 188: Start a new paragraph here (and everywhere else, where you start to discuss a new topic).

➢ Reply: We have changed accordingly.

Lines 104-105, 122, 133 (could be resulted), 216-220, 291, 303-304 and many other lines: Odd language

➢ Reply: We have changed these sentences to:

➢ 104-105: "how do aerosol feedbacks change meteorological variables? and how do current models differ in estimating these changes?"

➢ 122: "It is noticed that M6 predicts lower aerosol optical depth (AOD) than M7 (Gao et al., 2018a), which could partly explain the weaker ADRF estimated by M6. M6 uses an external assumption of aerosol mixing state which likely cause weaker absorption and ADRF in the atmosphere (Conant et al., 2015)."

➢ 133: "among models could be due to assumptions"

➢ 216-220: "Concentrations of sulfate and organic aerosol are generally underestimated by most of the participating models, and M4 overestimates the concentrations of organic aerosols (Gao et al., 2018a). These model errors were attributed to the missing multiphase oxidation mechanisms of SO2, and different secondary organic aerosol (SOA) formation mechanisms in these models (Gao et al., 2018a)."

➢ 291: "The spatial distributions of ADRF at the surface and inside the atmosphere inferred from multiple models are generally consistent, but the spatial distributions of ADRF at the TOA estimiated by these models greatly differ."

➢ 303-304: "The results indicate the important effect of aerosol mixing state on the estimates of ADRF and aerosol feedbacks, and BC exhibits large contribution to atmospheric heating although it accounts for a small share of mass concentration of PM2.5"

Line 122: 'an external assumption': About what?

➢ Reply: external assumption of aerosol mixing state. We have changed it in the revised manuscript.

Line 122: Why 'also'? Which other models use also the assumption of external mixing?

➢ Reply: We have deleted "also" here

Line 169: A word seems to missing here.

➢ Reply: We have changed the sentence to "Model evaluation of PM2.5 composition in Gao et al. (2018a) reveals that M4 overpredicts the concentrations of organic carbon, which could be one of the reasons for the higher estimated temperature reduction due to aerosols."

Line 225: A reference should be given here.

➢ Reply: We have added the following reference: RIEMS-Chem model (M5) (Han et al., 2010)

➢ Han, Z.: Direct radiative effect of aerosols over East Asia with a regional coupled climate/chemistry model, Meteorologische Zeitschrift, 19(3), pp.287-298, https://doi.org/10.1127/0941-2948/2010/0461, 2010.

Line 282: A reference should be given here.

➢ Reply: We have added the reference "Topic 3 of MICS-Asia III (Gao et al., 2018a) focuses on understanding how current online coupled air quality models perform in capturing extreme aerosol pollution event in northern China and how aerosols interact with radiation and weather."

References in Figure captions: 2018a or 2018b? Better mention the models instead of the institutions (eventually not necessary for all figures)

➢ Reply: We have changed 2018 to 2018a. We have added model names: "(M1: WRF-Chem, Pusan National University; M2: WRF-Chem, University of Iowa; M4: NU-WRF, NASA; M5: RIEMS-Chem, Institute of Atmospheric Physics; M6: RegCCMS, Nanjing University; M7: WRF-CMAQ, University of Tennessee; Gao et al., 2018a). "

Line 450: Use either complete list of authors (Forkel, R., Balzarini, A., Baró, R., Bianconi, R., Curci, G., Jiménez-Guerrero, P., Hirtl, M., Honzak, L., Lorenz, C., Im, U., Pérez, J. L., Pirovano, G., José, R. S., Tuccella, P., Werhahn, J., and Zabkar, R.: Analysis of the WRF-Chem contributions to AQMEII phase2 with respect to aerosol radiative feedbacks on meteorology and pollutant distributions, Atmos. Environ., 115, 630–645, 2015.) or 'et al' after the third author.

> ➢ Reply: Thank you for mentioning. This list was generated automatically with Mendeley. We have updated.